# Metallic Iron for Environmental Remediation: Starting an Overdue Progress in Knowledge

**Rui Hu** [1,*], **Huichen Yang** [2], **Ran Tao** [2], **Xuesong Cui** [1], **Minhui Xiao** [1], **Bernard Konadu Amoah** [1], **Viet Cao** [3], **Mesia Lufingo** [4], **Naomi Paloma Soppa-Sangue** [1,5], **Arnaud Igor Ndé-Tchoupé** [6], **Nadège Gatcha-Bandjun** [7], **Viviane Raïssa Sipowo-Tala** [5], **Willis Gwenzi** [8,*] and **Chicgoua Noubactep** [2,*]

[1] School of Earth Science and Engineering, Hohai University, Fo Cheng Xi Road 8, Nanjing 211100, China; cuixuesong@hhu.edu.cn (X.C.); xiaominhui@hhu.edu.cn (M.X.); bernardowuraku@yahoo.com (B.K.A.); soppanaomipaloma@yahoo.fr (N.P.S.-S.)

[2] Angewandte Geologie, Universität Göttingen, Goldschmidtstraße 3, D-37077 Göttingen, Germany; huichen.yang@geo.uni-goettingen.de (H.Y.); ran.tao@geo.uni-goettingen.de (R.T.)

[3] Hung Vuong University, Nguyen Tat Thanh Str., Viet Tri, Phu Tho 29000, Vietnam; caoviet@hvu.edu.vn

[4] Department of Water and Environmental Science and Engineering, Nelson Mandela African Institution of Science and Technology, Arusha P.O. Box 447, Tanzania; lufingom@nm-aist.ac.tz

[5] Faculty of Health Sciences, Campus of Banekane, Université des Montagnes, Bangangté P.O. Box 208, Cameroon; sipoworais@yahoo.fr

[6] Department of Chemistry, Faculty of Sciences, University of Douala, Douala B.P. 24157, Cameroon; ndetchoupe@gmail.com

[7] Faculty of Science, Department of Chemistry, University of Maroua, Maroua BP 46, Cameroon; nadegegatcha@yahoo.fr

[8] Biosystems and Environmental Engineering Research Group, Department of Soil Science and Agricultural Engineering, University of Zimbabwe, Mt. Pleasant, Harare P.O. Box MP167, Zimbabwe

[*] Correspondence: rhu@hhu.edu.cn (R.H.); wgwenzi@yahoo.co.uk or wgwenzi@agric.uz.ac.zw (W.G.); cnoubac@gwdg.de (C.N.)

**Abstract:** A critical survey of the abundant literature on environmental remediation and water treatment using metallic iron ($Fe^0$) as reactive agent raises two major concerns: (i) the peculiar properties of the used materials are not properly considered and characterized, and, (ii) the literature review in individual publications is very selective, thereby excluding some fundamental principles. $Fe^0$ specimens for water treatment are typically small in size. Before the advent of this technology and its application for environmental remediation, such small $Fe^0$ particles have never been allowed to freely corrode for the long-term spanning several years. As concerning the selective literature review, the root cause is that $Fe^0$ was considered as a (strong) reducing agent under environmental conditions. Subsequent interpretation of research results was mainly directed at supporting this mistaken view. The net result is that, within three decades, the $Fe^0$ research community has developed itself to a sort of modern knowledge system. This communication is a further attempt to bring $Fe^0$ research back to the highway of mainstream corrosion science, where the fundamentals of $Fe^0$ technology are rooted. The inherent errors of selected approaches, currently considered as countermeasures to address the inherent limitations of the $Fe^0$ technology are demonstrated. The misuse of the terms "reactivity", and "efficiency", and adsorption kinetics and isotherm models for $Fe^0$ systems is also elucidated. The immense importance of $Fe^0/H_2O$ systems in solving the long-lasting issue of universal safe drinking water provision and wastewater treatment calls for a science-based system design.

**Keywords:** adsorption capacity; decentralized water supply; electrochemical reaction; inconsistent view; sand filtration; wastewater treatment; zero-valent iron

## 1. Introduction

Metal corrosion is one of the most important problems in industry, transport and agriculture [1–4]. Understanding metal corrosion comprises its detection (e.g., analytical, visual), its monitoring (e.g., mass loss, $H_2$ evolution) and its long-term characterization under various field conditions [4,5]. The corrosion research aims at determining the durability of metallic structures under operational conditions (e.g., oil and gas pipelines, tanks) and revealing the mechanisms of corrosion process [5,6]. This mechanism can be chemical, electrochemical or mixed [1]. Various tools have been used to characterize the corrosion resistance of different metals under various application conditions [4]. The overall result is the availability of integrated approaches to assess and predict the corrosion processes and thus the longevity of metallic structures (e.g., buried pipes) [3,4]. However, the frequency and sudden nature of metallic pipe failures worldwide indicate the inadequacy of current knowledge related to the longevity of buried metallic pipes.

Metallic iron ($Fe^0$) used as a reactive material in subsurface permeable reactive barriers is comparable to iron pipes with three major differences: (i) corrosion is welcome because it is a rather useful process [7–9], (ii) a reactive wall is ideally permanently water saturated, and (iii) the length of used particles (<5 cm) is tiny compared to pipes which are up to 12 m in length. On the one hand, $Fe^0$ specimens used in water treatment comprise steel wool with thickness varying between 25 and 90 μm [10,11]. On the other hand, the length of these particles is comparable to the wall thickness of iron pipes (2–4 mm). There has been no real system analysis for remediation $Fe^0$ materials with the aim to outline the differences making their peculiar characteristics. In addition, traceably deriving the longevity of remediation $Fe^0$ specimens from $Fe^0$ pipes is impossible because of the differences highlighted.

The remediation $Fe^0$ was termed as zero-valent iron (ZVI). This acronym is perhaps the first problem of this still innovative technology. A literature research with "zero-valent iron" as keyword would never reveal the ancient literature on $Fe^0$ for water treatment [12–16]. Indeed, no research group until 2017 has referenced a single article from the ancient use of $Fe^0$ in water treatment [17,18]. In chemistry, metallic elements are characterized by their oxidation state, the one of $Fe^0$ is zero (0). Upon oxidation, $Fe^0$ is transformed to $Fe^{II}$, $Fe^{III}$ or $Fe^{IV}$ species. Under environmental conditions, only $Fe^{II}$ and $Fe^{III}$ species are stable. The oxidation of $Fe^0$ to $Fe^{II}$ is a redox process characterized by an electrode potential whose value is ™0.44 V [1]. According to the first principle of chemical thermodynamics, $Fe^0$ can be oxidized by oxidizing agents from each redox couple having a higher electrode potential ($E^0$ > ™0.44 V). Water ($H_2O$ or $H^+$) is a relevant oxidizing agent for $Fe^0$ under environmental conditions. The electrode potential for the redox couple $H^+/H_2$ is 0.00 V. $Fe^0$ immersed in (contaminated or polluted) water is corroded to form $H/H_2$ and $Fe^{II}$ (and mixed $Fe^{II}/Fe^{III}$) species which are stand-alone reducing agents [19–26]. Clearly, it is not surprising that selected species undergo reductive transformations in an $Fe^0/H_2O$ system [19,26–29]. The application of metallic iron in water treatment dates back to the 1890s (Table 1). Suspended matter then left to settle

**Table 1.** Historical application of metallic iron in water treatment (modified from Mwakabona et al. [17]).

| Age/Time | Material | Usage | Target Impurities | Application | Reference |
|---|---|---|---|---|---|
| Pre–1850 | Old iron nails | Shaken with impure water | Suspended matter then left to settle | Used in West England | [30,31] |
| 1850–1900 | Iron plates/wires (1857) | Suspended in flowing impure | Suspended matter water | Tested but not widely applied | [13,31,32] |
| | Magnetic carbide/ Iron oxide | Filter media | Suspended matter/microbes | Widely applied household scale filters | [12,13,31] |

**Table 1.** *Cont.*

| Age/Time | Material | Usage | Target Impurities | Application | Reference |
|---|---|---|---|---|---|
| | Polarite | Filter media | Suspended matter/colour/microbes | Household filters | [32,33] |
| | Spongy iron | Filter media | Suspended matter/colour/microbes/chemical contaminants | Household and large scale filters | [30,31,34,35] |
| | Iron fillings | Agitated in the evolving purifier | Suspended matter/colour/microbes | Large scale in treatment plants | [31,34,35] |
| 1900–1950 | Iron fillings | Agitated in the revolving purifier | Suspended matter/colour/microbes/chemical contaminants | Large scale in treatment plants | [16,31,35,36] |
| 1950–1990 | Steel wool | Filter media | Radionuclides | Widely tested household scale filters | [37] |
| | Iron fillings | Filter media | Selenium from agricultural drainage water | Field scale | [38] |
| | Iron fillings | Static or dynamic | Halogenated hydrocarbons | Concept | [39] |
| Post–1990s | Iron fillings, steel wool | Filter media | Phosphate from agricultural drainage water | Field scale | [40,41] |
| | Iron fillings | Filter media | Domestic wastewater | Field scale | [42] |
| | $Al^0/Fe^0$ composite | Filter media or batch systems | Pathogen and chemical removal | Laboratory demonstration | [43] |
| | Iron fillings | Filter media | Pathogen removal | Laboratory demonstration | [44] |
| | Iron nails | Filter media | Kanchan Arsenic filter | Field scale | [45,46] |
| | Composite iron matrix | Filter media | SONO Arsenic filter | Field scale | [47] |
| | Metallic iron | Reactive media in PRBs | Chemical contaminants | Field scale | [22,23,48,49] |
| | Metallic iron | Filter media or batch systems | All classes of contaminants | Concept | [50] |
| | Metallic iron | Filter media | *E. coli* and *Listeria monocytogenes* removal from surface water for irrigation | Field scale | [51,52] |

Table 1 presents a summary of the historical applications of metallic iron in water treatment. The history of $Fe^0$ application in environmental remediation has been the subject of several papers by our research group. Comprehensive research on $Fe^0$ for water treatment revealed that the generation of iron hydroxides and oxides (iron corrosion products or FeCPs) is the root cause of contaminant removal in $Fe^0/H_2O$ systems [22,23,25,26,53,54]. For example, arsenic [55], carbon tetrachloride [29], chromium [56], fluoride [57,58], hexachloroethane [59] , methylene blue [60], methyl orange [61], Orange II [62], phosphate [63], selenium [64] and zinc [65] are all removed in $Fe^0/H_2O$ systems despite their differences in charge, redox-reactivity and size.

A critical research review article from 2008 [54] established that adsorption, co-precipitation and size-exclusion were the fundamental mechanisms of contaminant removal in $Fe^0/H_2O$ systems. Yet to date, 13 years later, researchers are still trying to establish the mechanisms by which aqueous contaminants are removed in the presence of $Fe^0$ [59,63,64,66,67]. Unfortunately, none of them has proven the alternative concept [53,54,68] wrong, and most of them are considering $Fe^0$ as the electron donor for observed transformations (electrochemical reaction) [69–71]. Moreover, "contaminant

reduction" and "contaminant removal" are mostly randomly interchanged. Therefore, a better argumentation is still needed to convincingly explain the in-depth knowledge on the mechanisms causing contaminant removal in $Fe^0/H_2O$ systems. Another important application of $Fe^0$ materials entails using $Fe^0$ to induce a pH shift and/or enhance microbial processes involved in methane production in anaerobic digesters [72–75]. This aspect is not considered herein as it is not focused on contaminant removal as discussed in Section 2.

This paper presents a profound analysis of the $Fe^0/H_2O$ system and derives the leading causes and factors influencing its efficiency for water treatment. Relevant factors include: (i) the $Fe^0$ specimen including its form and size (intrinsic reactivity), (ii) the water chemistry including the nature of contaminants, the presence of dissolved $O_2$ and the pH value, (iii) the contact time (flow velocity or mixing intensity), and (iv) the $Fe^0$ amount and its proportion in the reactive mixture (thickness of the reactive layer or number of columns). Contaminants are explicitly considered within water chemistry, one of the four main groups of factors influencing iron corrosion. The approaches conventionally adopted to investigate these factors, the major findings, the limitations and the knowledge gaps are presented. It is argued that no progress in knowledge is possible before the research community agrees on the key issue that $Fe^0$ is not a reducing agent (under field conditions).

## 2. The $Fe^0/H_2O$ System

### 2.1. Overview of Fundamental Aspects

There is a transfer of electrons from the $Fe^0$ body (solid state) to the $Fe^0/H_2O$ interface whenever a piece of a reactive $Fe^0$ specimen is immersed in an aqueous solution ($Fe^0/H_2O$ system) [1,5,6,76]. This occurs because $Fe^0$ is not stable under environmental conditions or because the redox couple $H^+/H_2$ ($E^0 = 0.00$) is higher than that of $Fe^{II}/Fe^0$ ($E^0 = ™0.44$) in the electrochemical series [1,22–26]. Equation (1) reveals that the oxidative dissolution of $Fe^0$ by protons ($H^+$) and Equation (1a) considers that protons are from water ($H_2O \Leftrightarrow H^+ + HO^-$). $Fe(OH)_2$ from Equation (1b) tends to polymerize and precipitate but can also be oxidized to even lower soluble $Fe(OH)_3$ by dissolved $O_2$ for example (Equation (2a)). $Fe(OH)_2$ and $Fe(OH)_3$ are polymerized and further transformed to various hydroxides and oxides (FeCPs) (Equation (3)) [1,21,77]. The different iron corrosion products (FeCPs) depict different adsorptive affinities for dissolved species [77–79]. Equation (4) summarizes the process of aqueous iron corrosion.

$$Fe^0 + 2\,H^+ \Rightarrow Fe^{2+} + H^2 \tag{1a}$$

$$Fe^0 + 2\,H_2O \Rightarrow Fe(OH)_2 + H_2 \tag{1b}$$

$$4\,Fe^{2+} + O_2 + 4\,H^+ \Rightarrow 4\,Fe^{3+} + 2\,H_2O \tag{2a}$$

$$4\,Fe(OH)_2 + O_2 + 2\,H_2O \Rightarrow 4\,Fe(OH)_3 \tag{2b}$$

$$Fe(OH)_2,\ Fe(OH)_3 \Rightarrow FeO,\ Fe_3O_4,\ Fe_2O_3,\ FeOOH \tag{3}$$

$$Fe^0 + H_2O + (O_2) \Rightarrow H_2 + \text{iron hydroxides and oxides} \tag{4}$$

In summary, Equation (4) recalls that immersing a reactive $Fe^0$ in water can be universally used to generate $H_2$, $Fe^{2+}$ and various $Fe^{II}$, $Fe^{III}$ and $Fe^{II}/Fe^{III}$ hydroxides and oxides. Equation (1) demonstrates that $Fe^0$ is a scavenger of humidity ($H_2O$), while Equation (2) demonstrates the $O_2$ scavenging nature of $Fe^0$. These two scavenging characteristics have been exploited in several industrial applications [80–82]. For example, $Fe^0$ is used as desiccant in food packaging [81]. In the $Fe^0$ remediation literature, the best illustration for the nature of the $Fe^0/H_2O$ system as generator of FeCPs is perhaps the excellent research article by Furukawa et al. [83]. These authors used several analytical tools to demonstrate the presence of ferrihydrite, green rust, magnetite and lepidocrocite in an $Fe^0/H_2O$ system. More importantly, they conclude that an $Fe^0/H_2O$ system is a temporally and spatially heterogeneous geochemical environment. Concerning the spatial heterogeneity, Furukawa et al. [83] specified that magnetite ($Fe_3O_4$) is generated

in the vicinity of $Fe^0$, whereas ferrihydrite ($Fe(OH)_3$) precipitates away from the $Fe^0$ surface. This conclusion corroborates ancient findings [84] and recalls that even under oxic external conditions, there is a progressive $O_2$ depletion culminating into anoxic conditions in the vicinity of $Fe^0$. In this context, Stratmann and Müller [85] clearly demonstrated that oxygen is reduced by $Fe^{II}$ species within the oxide scale (chemical reaction), while $Fe^0$ is oxidized by water (electrochemical reaction).

This section has recalled that, before $Fe^0$ complete depletion, an $Fe^0/H_2O$ system is a dynamic and heterogeneous system containing $Fe^0$ and all its corrosion products (Equation (4)). Accordingly, even the most accurate measurements and the most precise observations are just a static snap-shots of dynamic processes within the system. The situation is exacerbated by the evidence that, the processes occur over an enormous range of timescales ranging from some few minutes or hours in laboratory investigations to months and years in field applications [86–88]. This communication insists on the fact that the frequency of discrepant reports in the scientific literature is rooted on an insufficient system analysis. In this regard, an accurate system analysis constitutes the theory of the system. The theory of the $Fe^0/H_2O$ system, in turn, is like a guide to constrain the choice of the model. The theory that some contaminants are reduced by electrons from $Fe^0$ (electrochemical mechanism) is flawed as water, even present as humidity or moisture do corrode iron [1]. In fact, even deionized water corrodes iron [89]. Figure 1 depicts the interactions of contaminants with solid phases in a pure adsorbent system and the $Fe^0$-based system.

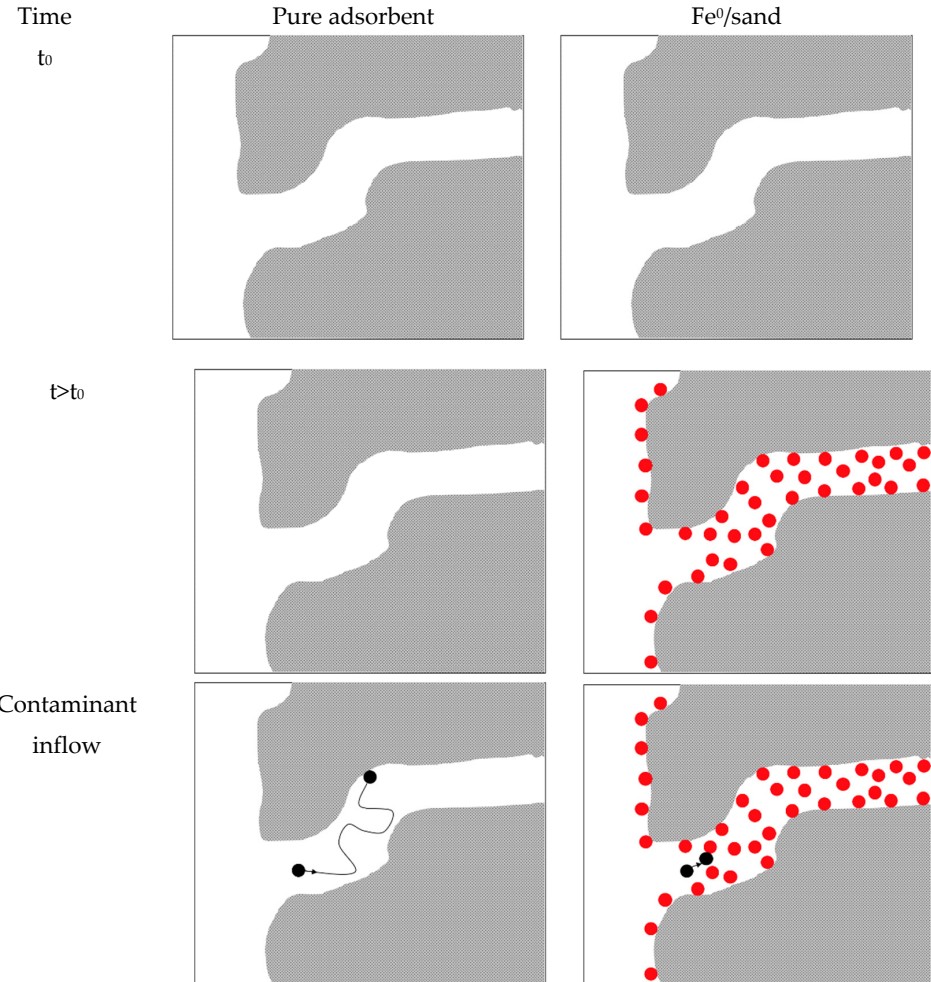

**Figure 1.** Schematic diagram comparing the interactions of contaminants (black points) with solid phases in a pure adsorbent system (left) and the $Fe^0$-based system (right). The red points represent iron corrosion products (FeCPs) which are either coated on solids or suspended in the pore solution.

## 2.2. Oxide Scale on $Fe^0$ and the Decontamination Process

The universal oxide scale on iron metal (at pH > 4.5) is still regarded by the majority of active researchers on remediation $Fe^0$ as a disturbing factor compromising the electron transfer from the metal body (reactivity loss) [90–92]. This view contradicts the evidence that a lag time between the start of experiments and reductive transformation is reported in the literature [93,94]. This time corresponds to the quantitative generation of FeCPs, which act as contaminant scavengers. This means that relevant reducing agents are generated in situ. Another problem of the $Fe^0$ literature is that "contaminant removal" and "contaminant reduction" are randomly interchanged, while no real mass balance of the contaminants has been presented [53,54,95]. On the contrary, contaminants that are not recovered are assumed to be chemically reduced [68–70].

A look at the mechanism of oxide scale formation reveals that it cannot be electronically conductive. In fact, the initial scale is very porous and cannot transfer electrons because air and water are not electronically conductive (an aqueous solution can be ionic conductive—electrolyte). In subsequent stages, available pores are filled with nascent FeCPs, but they are never uniform and the oxide scale is a mixture of iron hydroxides and oxides [5]. An oxide scale made up of $Fe_3O_4$ alone would have been electronically conductive. However, such an $Fe_3O_4$ scale cannot exist under natural conditions (immersed $Fe^0$). All other FeCPs are at best semi-conductors and cannot relay electrons from $Fe^0$ under natural conditions. Clearly, reports justifying the reductive efficiency of $Fe^0/H_2O$ systems using the semi-conductive nature of FeCPs are mistaken [96]. This assertion encompasses the $Fe^0/pyrite/H_2O$ system, whose efficiency is mainly justified by the semi-conductive nature of FeS species [97–99].

The oxide scale on $Fe^0$ is definitively a diffusion barrier for all dissolved species, including the pollutants. It is also the contaminant scavenger such that electrochemical corrosion of immersed $Fe^0$ induces the generation of contaminant scavengers and other reducing agents. Thus the generation of solid FeCPs is a necessary process which has the (perceived negative) side effect of being expansive. Thus, designing an efficient and sustainable system requires answering the question: how long can FeCPs be generated to satisfactorily treat water while keeping a reasonable hydraulic conductivity (permeability)?

## 2.3. Chemical Aspects

As discussed earlier (Section 2.1), $Fe^{2+}$ from Equation (1) is transformed to ferrous hydroxide $(Fe(OH)_2)$ and ferric hydroxide $(Fe(OH)_3)$ which have a strong tendency to form colloids of particles that normally carry a positive charge [62,100,101]. These minerals are further transformed to other $Fe^{II}/Fe^{III}$ minerals (e.g., $Fe_2O_3$, FeOOH, green rust) exhibiting different affinities to dissolved species. The nature of oxides in each individual system depends on the intrinsic reactivity of the used $Fe^0$ material and the environmental conditions [5,79]. For example, two different $Fe^0$ specimens corroding under the same environment will not necessarily produce the same iron oxides, because the composition of the oxide scale depends on the relative kinetics of $Fe^0$ dissolution and Fe hydroxide precipitation, which in turn depends on the solution chemistry, including the pH value, dissolved ions and the salinity [5,6,102]. On the other hand, in-situ generated free $Fe^{2+}$ are adsorbed to the surface of available minerals to form the so-called structural $Fe^{II}$ with a reducing power far larger than that of the free $Fe^{2+}$ ($E^0 < 0.77$ V) and sometimes stronger than $Fe^0$ ($E^0 < –0.44$ V) [103]. The availability of several reducing agents in the $Fe^0/H_2O$ system, and especially from structural $Fe^{II}$, partly stronger than $Fe^0$ implies that the electrochemical series of metal alone cannot predict the chemistry of the system.

## 2.4. Physical Aspects

The volumetric expansive nature of iron corrosion is the most important physical phenomenon occurring in $Fe^0/H_2O$ systems [104]. There is expansion because the parent metal ($Fe^0$) produces in-situ both: (i) $H_2$, occupying a volume about 3100 times larger [26,105], and (ii) each solid oxide and hydroxide is at least twice larger in volume than $Fe^0$ ($V_{oxide} > V_{iron}$) [106,107]. For example, the specific

density of magnetite ($Fe_3O_4$) is about one half that of iron ($Fe^0$). It implies that after corrosion a space twice larger than the initial space is occupied [108–110]. While it can be assumed that $H_2$ escapes from each open system, no free expansion of oxides in porous systems (e.g., water filters, reactive walls) can be assumed [26]. External or internal free expansion occurs in metallic pipes [111,112] and on the walls of steel canister for radioactive waste repositories [110,113]. On the contrary, free expansion cannot be expected in steel-reinforced concrete structures [108,109]. Table 2 presents a comparison of iron corrosion parameters in $Fe^0$ remediation to that of water pipes and reinforced concrete. Accordingly, considering expansive iron corrosion, which culminates into permeability loss is an essential design parameter for porous $Fe^0/H_2O$ systems ($Fe^0$ filters). For each $Fe^0$ filter, the temporal production of both $H_2$ and oxide is decisive for the long-term efficiency and the permeability of the system [105,107,114,115].

**Table 2.** Comparison of iron corrosion parameters in $Fe^0$ remediation to that of water pipes and reinforced concrete.

| Parameter | Water Piping | Reinforced Concrete | $Fe^0$ Remediation |
|---|---|---|---|
| Iron corrosion | Destructive | Destructive | Constructive |
| Environment | Unknown soil | Known concrete | Known mixture |
| Fe size | several meters | some cm | <5 mm |
|  | some cm thick | some mm | <2 mm |
| Rust expansion | Minor or no issue | major issue | major issue |
| Corrosion effect | Pipe damage | cracking | permeability loss |
| Linear corrosion | Conservative | non applicable | Absurd |
| Knowledge status | Quite good | Quite good | Rather poor |
| Service life | >50 years | >50 years | Up to 30 years |
| Reference | Enning et al. [2] | Caré et al. [108] | Hu et al. [26] |

Another key feature in investigating the remediation $Fe^0/H_2O$ system is that, at pH > 4.5, the $Fe^0$ surface is permanently covered by an oxide scale. The oxide scale acts both as: (i) conduction barrier for electrons from the metal body, and (ii) physical barrier for dissolved species, including $O_2$ and pollutants of concern [22–26,53,54]. The net result is that $Fe^0$ is oxidized by water (electrochemical reaction—Equation (1)), and $O_2$ and dissolved contaminants are reduced by reducing species present in the oxide scale ($Fe^{II}$ and $Fe^{II}/Fe^{III}$ species, $H_2$) (chemical reaction). Again, any reasoning based on the electrochemical series of elements is a misuse of chemical thermodynamic, as the physics of the system, specifically the expansive nature of iron corrosion and it electronically non-conductive nature are simply ignored.

*2.5. Kinetic Aspects*

Aqueous corrosion of $Fe^0$ materials under environmental conditions (pH > 4.5) is an electrochemical process involving iron dissolution at the anode and $H_2$ evolution at the cathode (Equation (1)). This electrochemical reaction is accompanied by the formation of an oxide scale on $Fe^0$ which is not protective as a rule [6,116–118]. In general, oxide scale growth and its protectiveness depend primarily on the precipitation rate of iron hydroxides [5,102]. As the $Fe^0$ surface corrodes under the initial scale, corrosion continuously undermines the scale. Voids are created and are progressively filled up by the ongoing hydroxide precipitation. The relative rate of (i) $Fe^0$ oxidative dissolution and (ii) hydroxide precipitation in the $Fe^0$ vicinity determine the protectiveness of the oxide scale. According to Nesic [5], when the rate of hydroxide precipitation exceeds the rate of $Fe^0$ dissolution, a dense protective oxide scale is formed. Conversely, when $Fe^0$ dissolution undermines the new oxide scale faster than hydroxide precipitation can fill in the voids, a porous and non-protective scale forms.

There are several factors influencing the iron corrosion rate, including the intrinsic reactivity of a material and the solution chemistry. The solution chemistry includes the presence of dissolved $O_2$, contaminants and other (mostly) ubiquitous species. In particular, two different $Fe^0$ specimens may exhibit different degree of protectiveness under the same operational conditions. The most important feature to consider is that the corrosion rate is never linear. Thus, a linear extrapolation of the initial corrosion rates of $Fe^0$ specimens in engineered systems can give an inaccurate estimation of its service life. However, for remediation systems, the non-linear nature of the corrosion kinetics implies a decrease in efficiency. Again, results from pipe corrosion or wall corrosion cannot be transferred to remediation $Fe^0/H_2O$ systems. However, it is certain that the semi-permeable nature of the oxide scale allows significant and continuous corrosion over long time periods ("rust never rest"). In this context, Roh et al. [119] reported of buried iron pieces from World War I still corroding in the subsurface. Thus, the objective of the $Fe^0$ remediation is to couple this long-term corrosion with efficient contaminant removal [7,88,118,120,121].

## 2.6. Investigating the Fe⁰/H₂O System

The extent and kinetics of $Fe^0$ oxidative dissolution are the result of the following: (i) the nature of metal (intrinsic reactivity), and (ii) the interactions between $Fe^0$ and the environment in which it is placed [76,122–125]. Therefore, (i) a change of material, and/or (ii) a change in environment results in changes in the rate and the extent of corrosion. In this study, the influence of the environment is considered at micro-scale, specifically what is happening on the $Fe^0$ surface, in its vicinity or over short distances (within the oxide scale). The oxide/water interface is also considered, but the volume of the solution is not. The aqueous phase is regarded as a reservoir of pollutants and co-solutes, while being the principal $Fe^0$ oxidizing agent [126,127].

Laboratory and pilot-scale investigations are usually conducted in order to obtain reliable information on the interactions of metallic devices with particular operational environments [128,129]. Laboratory tests are designed to simulate some relevant field situations. Laboratory studies are mostly aimed at obtaining data in a more convenient way and in a shorter time [123,124,130]. They also provide mechanistic information for field applications. However, short-term laboratory experiments are always a simplification and this should be borne in mind when interpreting achieved results [76,129,131,132]. Given the diversity of operational parameters that have been proven to influence iron corrosion from individual studies, one can be overwhelmed by their number and the fact that each material is unique in its corrosion behaviour [11,124,125,133]. Therefore, a first attempt toward a systematic investigation of relevant influencing factors goes through the consideration of the electrochemical nature of aqueous iron corrosion.

$Fe^0$ is a good conductor of electricity, and its electrochemical aqueous corrosion depends on the conductive nature of the solution in which it is immersed [1,76]. Understanding the corrosion processes helps in selecting $Fe^0$ materials necessary for designing sustainable systems. Therefore, the best way to design a sustainable system is to consider the fundamentals of iron corrosion at the design stage. This section highlights the knowledge of corrosion principles and the importance of the environment and materials for field design. The four compartments necessary for continuous $Fe^0$ corrosion are: (i) an anodic region on $Fe^0$ where the metal is oxidized and releases $Fe^{2+}$ (leaving two electrons behind), (ii) an electrolyte to transport released $Fe^{2+}$ away from the anode, (iii) a cathode where the simultaneous reductive transformation coupled to iron oxidation occurs, and (iv) the $Fe^0$ body transporting electron from the anode to the cathode. In an $Fe^0/H_2O$ system, the anode and the cathode are different sites on the same $Fe^0$ specimen. In the conventional remediation technology, the size of $Fe^0$ particles is generally small (<5 mm). The four compartments must be electronically connected for the electrochemical process to proceed. This means that if $Fe^0$ is covered by an oxide scale, the scale must be conductive to warrant the transfer of electrons from the metal body to the oxidizing agent within the oxide scale or at the oxide/$H_2O$ interface.

A $Fe^0/H_2O$ system is made up of two interfaces: (i) $Fe^0$/oxide and (ii) oxide/$H_2O$. Because the oxide scale is never electronically conductive and is a diffusion barrier to many species, only water can quantitatively reach the $Fe^0$ surface. The net result is that $Fe^0$ oxidative dissolution is an electrochemical reaction (water is reduced) but all other observed/reported chemical reduction occur within the oxide film or at the oxide/$H_2O$ interface [53,54,127]. For the $Fe^0/H_2O$ remediation system, it means that contaminants are not reduced by electrons from $Fe^0$. This has important implications on the operating principles of $Fe^0/H_2O$ systems: first, using the electrochemical series to predict the reductive transformation of any species has been a mistake, and second, using the stoichiometry of any electrochemical reaction involving $Fe^0$ is also a mistake. Accordingly, contaminants are 'just' dissolved species, capable of modifying: (i) the conductive properties of the electrolyte ($H_2O$), (ii) the ion conductivity of the oxide scale, and (iii) formation and the transformation of the oxide scale.

## 3. An Overview of the Mistakes of Past Efforts

### 3.1. Contaminant Removal Mechanisms

The major mistake of the $Fe^0$ remediation literature has been to consider that relevant contaminants are reduced by an electrochemical reaction (electrons from the metal) [134–136]. This section summarizes the extent of the confusion using a paper by Kamolpornwijit and Liang [137]. This paper is selected because it considered past efforts in understanding the service life of $Fe^0$ filters and performed long-term experiments (>400 days). Kamolpornwijit and Liang [137] considered porosity loss during nitrate ($NO_3^-$) removal in an $Fe^0$ barrier. The following reactions were considered:

$$Fe^0 + 2\,H_2O \Rightarrow Fe^{2+} + H_2 + OH^- \tag{5}$$

$$4\,Fe^0 + NO_3^- + 7\,H_2O \Rightarrow 4\,Fe^{2+} + NH_4^+ + 10\,OH^- \tag{6}$$

$$5\,Fe^0 + 2\,NO_3^- + 6\,H_2O \Rightarrow 5\,Fe^{2+} + N_2 + 12\,OH^- \tag{7}$$

$$10\,Fe^{2+} + 2\,NO_3^- + 6\,H_2O \Rightarrow 10\,Fe^{3+} + N_2 + 12\,OH^- \tag{8}$$

Equation (5) corresponds to Equation (1) and represents aqueous iron corrosion, an electrochemical reaction. Direct electron transfer from $Fe^0$ to $NO_3^-$ yielding $NH_4^+$ (Equation (6)) or $N_2$ (Equation (7)) are only possible if the oxide scale on $Fe^0$ is electronically conductive. Because an electronically conductive oxide scale does not exist under immersed conditions, Equations (6) and (7) are wrong. In discussing their results, Kamolpornwijit and Liang [137] considered that the $H_2$ volume they measured corresponds to the stoichiometry of Equation (5). Comparing this $H_2$ volume to the $N_2$ volume after Equation (7), they concluded that the contribution of water to the corrosion of iron is minimal. In chemical terms, this is erroneous as Equation (5) shows that 5 moles of $Fe^0$ release only 1 mole of $N_2$. Whether $N_2$ escapes from the system or not, the 5 moles of $Fe^{2+}$ are further oxidized and or precipitated within the $Fe^0$ filter and filling the initial porosity. In reality, it is 10 moles of $Fe^0$ that are needed to release one mole of $N_2$ (Equation (8)).

One major problem of the $Fe^0$ literature has been the discussion of the reaction mechanism without mass balance considerations [31,95,138]. Complete mass balance analysis comprises the one of iron, which is admittedly a difficult task as the system is highly dynamic. However, the analysis made herein for nitrate reduction clearly demonstrates the wrongness of the view that $Fe^0$ is corroded by $NO_3^-$. On the other hand, the importance of FeCPs in inducing porosity loss is demonstrated. If the discussion of the porosity loss starts with the assumption that one mole of $N_2$ corresponds to 10 moles of corroded and expanded $Fe^0$, a better evaluation of changes in the porosity will be achieved. Kamolpornwijit and Liang [137] is also an excellent illustration on how in-situ generated FeCPs are not properly considered while exotic species like $CaCO_3$ or the flow regime (convection versus laminar) are given key role in pore filling and permeability loss. In essence, if a contaminated water is rich in carbonates a pre-treatment unit for their removal can be used such that decontamination units are

$HCO_3^-$ free. Thus, given the expansive nature of iron corrosion, loss of permeability and porosity even occurs in aqueous systems without $CaCO_3$.

Another important feature from Kamolpornwijit and Liang [137] is that, in case nitrate is reduced to $NH_4^+$, regardless of the real mechanisms, $NH_4^+$ must be removed from the aqueous phase. This removal occurs by adsorption, co-precipitation and size-exclusion [53,54]. These three mechanisms represent the fundamental mechanisms of contaminant removal in $Fe^0/H_2O$ systems [26,67]. It is essential to recall that chemical reduction and even chemical precipitation are not relevant contaminant removal mechanisms in the concentration ranges relevant to natural waters [111,139]. In particular, for safe drinking water provision, physical methods (e.g., adsorption, filtration, ion exchange) are always mandatory to cope with the stringent regulations. As an example, water defluoridation by chemical precipitation yields an equilibrium fluoride concentration of about 8 mg $L^{-1}$ according to the solubility of $CaF_2$ [57,58]. This value is far larger than the maximum permissible contamination level of 1.5 mg $L^{-1}$ for drinking water. Coming back to the $Fe^0/H_2O$ system, available results show limited removal extent for fluoride while various removal extents for several contaminants depicting no redox reactivity in the systems has been documented. For this reason, a more rational approach is to consider that all species can be removed regardless of their redox potential, and identify exceptions on a case by case basis. In this regard, recent results by Hildebrant et al. [140] have confirmed that fluoride removal is low, but less reactive materials (EDTA test) comparatively remove more fluoride. This last observation corroborates the view that there is no single $Fe^0$ material for all situations and calls for more systematic investigations to identify appropriate $Fe^0$ materials for specific remediation applications.

The lack of systematic investigations has also affected the selection of $Fe^0$ materials used for pathogen removal [141–146]. Using various $Fe^0$ materials and very different experimental conditions, discrepant results have been reported [141,146]. Lu et al. [142] can be regarded as a perfect reflection of the state-of-the-art knowledge. The same authors investigated the mechanism of nitrate reduction and iron cycling by an iron-reducing bacteria strain (strain CC76) and metallic iron. They reported that the strain CC76 was able to utilize $Fe^{2+}$ (from iron corrosion) as electron donor for the nitrate removal. More importantly, they observed that $Fe^0$ inhibited the growth of strain CC76 in the early stage of the operation. This observation corresponds to CC76 removal by $Fe^0$. However, the authors also reported that after the initial stage, strain CC76 was able "to tolerate" the presence of $Fe^0$, meaning that no or less removal was achieved. This phase corresponds to a decrease in the kinetics of iron corrosion as discussed above and has been described in microbially-influenced corrosion [2]. If we recall the work of Kamolpornwijit and Liang [137] primarily investigating nitrate removal in an abiotic $Fe^0/H_2O$ system, it then becomes apparently clear that the major source of discrepancy in the literature is the insufficient system analysis. Like with chemical contaminants, all pathogens will be removed in a well-designed system. For a filtration system, the key questions to address are: (i) which $Fe^0$ material, in what amount, and for which contaminant and water? and (ii) with which filtration depth, for which flow velocity and for which operational duration? Another key aspect to consider is the nature of the aggregates (e.g., gravel, $MnO_2$, pumice, sand) to be mixed with $Fe^0$ and the $Fe^0$ proportion in the mixture [107,114]. These aspects are yet to be addressed in the $Fe^0$ remediation literature, but are critical for the design and field application of the technology. Table 3 presents a synthesis of current knowledge including the mistakes, and proposed refinements for future studies on $Fe^0$ remediation.

**Table 3.** Summary of synthesis of current knowledge and proposed refinements for future studies on $Fe^0$ remediation.

| Previous Study | Synthesis | Refinement for Future Studies |
|---|---|---|
| | Electrochemical reduction | Chemical reduction |
| | Protons as concurrent $Fe^0$ oxidizer | Protons as sole oxidizing agents for $Fe^0$ |
| Decontamination mechanism | Adsorption as possible mechanism | Adsorption as fundamental mechanism |
| | Co-precipitation as possible mechanism | Co-precipitation as fundamental mechanism |
| | Size-exclusion rarely considered | Size-exclusion as fundamental mechanism |
| Driving force | $E_0$ value = −0.44 V | Affinity to iron oxides and hydroxides |
| Selectivity | $E_0$ value of the contaminant | Surface charge and size |
| | $H_2$ evolution | $Fe^0$ dissolution in 1,10-Phenanthroline |
| Material selection | Removal efficiency for selected species | $Fe^0$ dissolution in EDTA |
| | Physical characterization tools | Not really useful |
| | $Fe^0$ dissolution in complexing agents | Phen test as candidate for standard method |
| Batch experiments | Shaken or stirred | Quiescent or shaken at < 100 rpm |
| | Lasting for some hours to days | Lasting for some weeks to months |
| | Mostly lasting for some few months | Lasting for at least six months |
| | Many contain only $Fe^0$ | Never contain more than 50% (vol/vol) $Fe^0$ |
| Column/field experiments | Some accelerated column experiments | Never artificially accelerated iron corrosion |
| System modelling | Based on the stoichiometry of an electrochemical reduction reaction | Consider protons as the sole oxidizers of $Fe^0$ Consider porosity loss related to each $Fe^0$ atom |

*3.2. Reactivity and Efficiency*

In the $Fe^0$ remediation literature the terms "reactivity" and "efficiency" are mostly randomly interchanged. This has introduced confusion in the evaluation of independent results. In an effort to resolve this confusion the notion of "electron efficiency" has been introduced [147,148]. The electron efficiency (in %) is defined as the proportion of electrons from $Fe^0$ oxidation that is used for the target reduction reaction, for example for nitrate reduction by Kamolpornwijit and Liang [137] (Section 3.1). In this approach, electrons used to reduce water (Eq. 1) or dissolved $O_2$ are considered as "excess electrons" or avoidable electron wastage [148]. In other words, the main reaction is devalued to a side effect. Contrary to this still prevailing approach [92], the present work and related ones [149–153] recall that aqueous iron corrosion (Equation (1)) is not an unwanted reaction leading to an extra consumption of $Fe^0$, but Equation (1) produces $Fe^{II}$ species and $H_2$, which are responsible for the documented chemical reduction and FeCPs which are responsible for contaminant removal and permeability loss [105–107].

Each $Fe^0$ material is characterized by its intrinsic reactivity while each system is characterized by its efficiency for water treatment [154–156]. According to Miyajima and Noubactep [156], the efficiency is the expression of reactivity in a given system. This means that changing the $Fe^0$ material modifies the efficiency. Conversely, an $Fe^0$ material proven efficient in a system could be useless in another

system. In the quest of more efficient systems for water treatment, several $Fe^0$ materials and groups of materials (e.g., bimetallics, iron nails, nano-$Fe^0$, scrap iron, sponge iron, steel wool) have been tested and used for water treatment (Table 1). These efforts have been rendered difficult by a mistaken system analysis [157,158] and the evidence that most comparative works are based on testing materials for the removal of individual contaminants [159–162]. Lufingo et al. [11] recently discussed approaches that can be considered as universal as they are based on the stoichiometry of Equation (1): (i) $H_2$ evolution [163,164] and (ii) $Fe^{2+}$ production [11,159]. They concluded that iron dissolution in a dilute solution of 1,10-Phenanthroline (2 mM) (Phen test) is the best available tool to characterize the intrinsic reactivity of $Fe^0$ materials (Section 4).

### 3.3. Misuse of Adsorption Isotherms and Kinetic Models

Several studies investigating contaminant removal by $Fe^0$ materials have applied isotherm and kinetic models initially developed for materials where contaminant removal occurs via adsorption [131, 132,165,166]. A detailed discussion of the various adsorption isotherm and kinetic models, including assumptions and equations are presented in earlier reviews [132]. A comprehensive review of the limitations associated with the applications of such models to $Fe^0$ systems is the subject of another paper, hence is beyond the scope of the current study. Briefly, the extension of such models to $Fe^0$ materials is problematic for the following reasons: first, unlike adsorbents such as activated carbon whose surface area is readily available for adsorption at the beginning of the experiment (time $t_0$ = 0; Figure 1), $Fe^0$ systems are highly dynamic and equilibrium is rarely achieved in such systems (i.e., rust never rests), while the total reactive sites contributing to contaminant removal is not exactly known (Figure 1). Thus, most assumptions for such models are not valid for $Fe^0$ materials. Second, as discussed earlier, contaminant removal in $Fe^0$ systems does not occur solely via adsorption. In view of this, the application of adsorption models to $Fe^0$ systems constitute another fundamental error in $Fe^0$ literature. Yet despite a number of reviews highlight the mistakes, misuse and inconsistencies in the use of adsorption kinetic and isotherm models [53,54,67–71,131,132], the problem persists. Thus there is need to develop models for contaminant removal in $Fe^0$ systems that takes into account the iron corrosion phenomena and the fundamental mechanisms responsible for contaminant removal as highlighted in the current review.

Reference [157] severely questioned the validity of specific rate constants ($k_{SA}$) in $Fe^0/H_2O$ systems. These rate constants are based on the stoichiometry of contaminant reduction by electrons from the metal body ($Fe^0$) and are not considering the interactions of contaminants within the oxide scale [167]. As discussed in Section 2.2, the $Fe^0$ surface is not accessible to all contaminants. Even $k_{obs}$ values (which are normalized to the surface area to obtain $k_{SA}$) currently used in the literature are erroneous as they are rooted on the wrong reaction. However, the most severe mistake has been to use the adsorption capacity for $Fe^0$ which is considered a reducing agent. Moreover, an adsorption capacity is determined in a system where the reactive agent is not completely depleted.

## 4. Selection and Characterization of $Fe^0$ Materials

As discussed above, the scientific reason for using all $Fe^0$ materials (e.g., granular iron and bimetallics, iron filings, iron nails, iron wire, nano-$Fe^0$, scrap iron, steel wool) in water treatment is the electrode potential of the redox electrode $Fe^{II}/Fe^0$: $E^0 = -0.44$ V. Clearly, all reactive $Fe^0$ materials have the same redox potential. The intrinsic reactivity of individual $Fe^0$ specimens depends on a myriad of factors from which some are not readily accessible to the researcher. Relevant influencing factors include: alloying elements, $Fe^0$ form, manufacturing processes, metallography, $Fe^0$ grain size, surface area, and surface oxidation state. Accordingly, each $Fe^0$ material has its own intrinsic reactivity which should be characterized in order to better understand how it is influenced by operational conditions to induce the intended remediation goal.

The long history of using $Fe^0$ for technical chemical applications, including water treatment reveals that there has always been efforts to select appropriate materials for individual applications.

For example, the porous "spongy iron" (sponge iron or direct reduced iron) was proven more suitable in filtration systems than dense materials [12,13,168]. Similarly, multi-metallic systems and nanoscale materials were recently developed to address (recalcitrant) contaminants that were less sensitive to treatment with granular materials [169]. For completeness, it should be stated that there is no $Fe^0$ material for all situations such that one should rationally select appropriate materials for each specific application. For example, while treating water in fluidized systems (Anderson Process), dense materials were better than sponge iron [13,168]. Similarly, Hildebrant et al. [140] reported that $Fe^0$ materials of low reactivity according to the EDTA test exhibited a better efficiency for fluoride removal. The question arises how to select the right $Fe^0$ specimen for a given application? This calls for the development of standardized protocols for selection and characterization of $Fe^0$ materials for various applications.

It is unfortunate that material selection has not received the due attention given its central role for the technology. Lufingo et al. [11] recently gave an overview of the available tools for the characterization of the intrinsic reactivity of $Fe^0$ specimens. These authors based their work on an excellent review article by Li et al. [162] and insisted on the quantification of $Fe^{2+}$ from Equation (1). According to Lufingo et al. [11], quantifying $Fe^{2+}$ or $H_2$ evolution from Equation (1) are the best tools to characterize the intrinsic reactivity. However, quantifying both $Fe^{2+}$ and $H_2$ in natural systems suffers from the high reactivity of those primary corrosion products within the system, including their adsorption onto FeCPs and their action as own reducing agents. Clearly, measured amounts of $Fe^{2+}$ and $H_2$ represent an excess quantity and cannot be strictly used to quantify iron corrosion (at pH values >4.5). These considerations clearly show that the $H_2$ evolution method [163,164] is an approximation, while the EDTA method [170] is disturbed by dissolved $O_2$. All other methods are contaminant-specific, and thus of low value [133,160,161]. Lufingo et al. [11] then proposed iron dissolution in a dilute (2 mM) 1,10-Phenanthroline solution (Phen test) as a facile method free from the inherent shortcomings of all available methods.

In terms of affordability and applicability, the Phen test is currently the best available method to characterize the intrinsic reactivity of $Fe^0$ specimens. The test lasts for less than 36 hours and characterizes the initial kinetics of $Fe^0$ dissolution. It is suggested as a candidate for a standard method and is immediately useful for material selection (screening) and quality control [11]. Each $Fe^0$ specimen is characterized by its $k_{Phen}$ value which reflects its initial dissolution at the pH value of natural waters but without the interaction of the oxide scale. The $k_{Phen}$ values for nine steel wool specimens ($Fe^0$ SW) presented by Lufingo et al. [11] were such that $0.07 \leq k_{Phen}$ ($\mu g\ h^{-1}$) $\leq 1.30$. This shows a ratio of reactivity of 18.5 for $Fe^0$ SW specimens which are often tested as a uniform class of materials compared to granular materials for example. Hildebrandt et al. [10,140] presented the $k_{EDTA}$ values for 13 reactive $Fe^0$ SW and one granular $Fe^0$ as $3.7 \leq k_{EDTA}$ ($\mu g\ h^{-1}$) $\leq 130.8$. The lowest $k_{EDTA}$ value corresponds to granular $Fe^0$, suggesting a reactivity ratio of up to 36 between $Fe^0$ SW and granular $Fe^0$. No data comparing $k_{Phen}$ values for $Fe^0$ SW and granular $Fe^0$ are yet available. However, it is certain that the Phen test provides a confidence $Fe^0$ screening tool and is non-contaminant-specific. Therefore, if a significant body of $k_{Phen}$ data and data for characterized $Fe^0$ specimens for the removal of selected representative contaminants (or contaminant groups) are made available, then site-specific treatability studies would then be required only to fine-tune design criteria for the optimal performance of remediation $Fe^0/H_2O$ systems.

Our research group has long recognized the need for systematic characterization of $Fe^0$ materials in terms of their intrinsic reactivity and efficiency. The idea followed by this research group as summarized herein is to characterize the $Fe^0$ reactivity and the efficiency of $Fe^0/H_2O$ systems in a pollutant-independent-manner. The alternative is to agree on probing pollutants while using standard protocols. A reference $Fe^0$ material would also be necessary. The use of EDTA and Phen tests to characterize the intrinsic reactivity has already been discussed. The efficiency of the $Fe^0/H_2O$ system is characterized using the methylene blue (MB) discoloration method (MB method) [60]. Herein, the low affinity of MB for FeCPs is used to trace the abundance of in-situ generated iron oxides in comparatively

long-term experiments [62,101,159,171]. The Phen test provides a reliable guidance in selecting $Fe^0$ from the large catalog of available materials and to control the quality of newly manufactured ones. On the other hand, the MB method provides a reliable guidance for the characterization of the efficiency of $Fe^0/H_2O$ systems. Selected probing agents enable the discussion of the results, and orange II, methyl orange and reactive red 120 were positively tested in this regard [61,62]. Recent results by Hildebrandt et al. [140] suggest that fluoride could be the next candidate for affordable probing reagent. In fact, it was found that fluoride removal is more efficient by low reactive $Fe^0$ specimens.

## 5. Future Perspectives and Potential Applications

### 5.1. Investigating the Fe$^0$/H$_2$O System

The findings on which future research should be rooted must be based on the evidence that the oxide scale on iron is a diffusion barrier and shall never been disturbed in ways that are not reproduced under field situations [22,23,53,54,56,124,126,127,171–175]. This has been the motivation for adopting quiescent bath experiments as a more suitable approach to investigate processes occurring in $Fe^0$ permeable reactive barriers some two decades ago [20]. Then and now, most research groups consider the shaking intensity as a relevant operational parameter to be investigated almost in all instances without a quiescent system as a reference. However, there is no given convincing reason for the chosen mixing intensities tested in these experiments [124,125].

In 2005, Devlin and Allin [173] designed a glass-encased magnet batch reactor (GEM reactor) to investigate the impacts of selected anions on the efficiency of granular $Fe^0$ in removing aqueous contaminants using 4-chloronitrobenzene as a probe molecule. This design aimed to achieve a better comparability between results of different experiments by fixing the stirring method and the stirring rates [169,173]. Using this approach, all experiments should be conducted in the GEM reactor to ensure that the granular iron remains stationary while the solution is stirred. Noubactep et al. [124,127] later demonstrated that, while using a rotary shaker, the shaking intensity should never be larger than 100 rpm. Given that a stirring device (including the GEM reactor) can be difficult to acquire by low-income laboratories, quiescent experiments can be adopted as a rule [171].

Quiescent batch experiments have also been adopted for the determination of the initial corrosion rate of $Fe^0$ in EDTA ($k_{EDTA}$) and Phen ($k_{Phen}$). Herein, the experiments are stopped before solution saturation, meaning that $Fe^0$ specimens are characterized under conditions where no oxide scale is available. Contrary to the prevalent approach [152–164], $Fe^0$ specimens are not characterized for any contaminant removal efficiency [176,177]. Solution saturation corresponds to [Fe] values equivalent to the stoichiometry of iron complexation (e.g.,[Fe] = 112 mg $L^{-1}$ for 2 mM EDTA). Lufingo et al. [11,120,121] recently compared the EDTA and the Phen test and established the superiority of the Phen test, which is additionally more affordable.

Another key feature of the remediation $Fe^0/H_2O$ system is related to the oxide scale. The omnipresent oxide scale on $Fe^0$ is positively charged under natural conditions (pH > 5.0) [62,100,101,178]. Thus, the $Fe^0/H_2O$ system is an ion-selective one and preferentially removes negatively charged species like bacteria [142,146]. One original idea has been to characterize the discoloration of methylene blue (MB) by $Fe^0$/sand/$H_2O$ systems (MB method). The MB method is grounded on an historical work by Mitchell et al. [178], who observed that sand adsorbs less MB when it is coated with iron oxide. Thus, mixing the same mass of different $Fe^0$ specimens with the given mass of a sand specimen, and allowing them to equilibrate for the long-time in a MB solution enable the differentiation of the $Fe^0$ reactivity of various materials [62,101,176,177]. As a rule, the most reactive material produces the largest amount of iron hydroxides, which, in turn, in-situ coat sand and thus discolors MB the least. MB is thus not a model contaminant, but an operational tracer [171,177]. To ease the interpretation of results, methylene orange (MO) [61,176] or Orange II [62,101] can be used as anionic dyes since they have molecular sizes similar to that of MB. Phukan et al. [62,101,176]

additionally used reactive red 120 which is also an anionic dye, but is much larger in size that MO and Orange II.

One key advantage of the MB method is that its enables a visual observation of preferential flow within a reasonable time scale (some weeks) [62]. In designing an $Fe^0$-based filtration systems, a decrease in hydraulic conductivity (permeability loss) is expected, and an early contaminant breakthrough will be observed due to preferential flow [179–183]. Mineral precipitation is particularly intense in the entrance zone of the filter [181], but the created preferential pathways are extended throughout the whole water column [179,181]. The investigation of the process of creation and extension of preferential pathways has been analytically very challenging [181–183]. On the basis of observations using the MB method [176,177], changes generated in the entrance zone can be better followed and considered in modelling efforts. This last aspect is crucial for the development of the technology as it has been convincingly demonstrated that models currently predicting the service life of $Fe^0$ filters are rooted on a wrong premise [114,115].

*5.2. Designing the Next Generation Filter*

The determination of the amount of a given $Fe^0$ which long-term corrosion kinetics would allow a designed $Fe^0$/aggregate system to satisfactorily treat a given water for a certain time frame (e.g., 12 months) can be regarded as a routine work. This routine engineering application is however complicated by the complexity of the iron corrosion process and mistakes in past research as outlined herein. One major thinking mistake has been to consider that admixing $Fe^0$ with sand (the most used aggregate) would alter the decontamination kinetics [90,183,184]. Thus, admixing with sand has been mainly considered as an economic tool to safe $Fe^0$ costs and the negative effects on the resulting system discussed [185]. It has been recently demonstrated that only hybrid $Fe^0$ systems are sustainable [105–107].

Several hybrid systems have been successfully tested for water treatment including $Fe^0$/activated carbon [184], $Fe^0/Fe_3O_4$ [185], $Fe^0/MnO_2$ [186], $Fe^0$/pyrite [99] and $Fe^0$/sand [187]. While inert sand alone has clearly improved the efficiency of $Fe^0$ systems [188], the efficiency of other tested aggregates was attributed to the specific materials. The $Fe^0$/sand is regarded as in-situ sand coating [141]. It is certain that other materials will be similarly covered by FeCPs. Thus, it is questionable whether the systems really operate as described.

Rephrasing Notter [189], the success of an $Fe^0$ filter depends on four main factors: (i) the quantity and quality of the water to be produced (e.g., daily), (ii) the intrinsic reactivity of used $Fe^0$, (iii) the nature of the contaminant(s), and (iv) the availability of $Fe^0$ material (to renew exhausted systems). This key principle remains unchanged, one century later. All is needed are systematic investigations take each $Fe^0$ and each water source as a stand-alone design parameter.

The laboratory procedures used to characterize and evaluate $Fe^0$-based systems in batch and column experiments vary considerably among studies. For example, some studies use agitated batch experiments to investigate contaminant removal by $Fe^0$ [126,127], while others used batch experiments operated in quiescent mode, which is closer to field conditions [61]. This makes direct comparison of results among studies problematic, and could lead to misleading conclusions about the performance of $Fe^0$ materials. Therefore, there is need to develop standardized protocols for the evaluation of $Fe^0$ material in both batch and column experiments. In the case of batch experiments, such protocols should include specifying the particle/grain size, tests for determination of material reactivity (e.g., EDTA, Phen tests), liquid/solid ratio, sampling frequency and duration of experiment. For column experiments, such protocols should specify filter depth, duration of experiment, grain size of filter material and chemical properties of test solution to be used, while accounting for potential interference among solutes. Besides developing dedicated pristine $Fe^0$ materials for the water treatment industry, scope also exists to use $Fe^0$ material generated as wastes from other industries. Similarly, iron oxide-rich spent sludge from $Fe^0$-based water treatment systems can be used as raw materials in other industries such as the production of pigments (e.g., iron oxide red) [190], and even as filter material in the construction

industry. This cyclic flow of iron materials between the water treatment industry and other industrial processes could form part of a circular economy. Filter wastes may also be regenerated and recycled to new $Fe^0$ material, and then used in other industry when they contain toxic contaminants such as As or U. The recycling of filter wastes in other industries may require detailed environmental risk assessments, including the evaluation of contaminant leaching and potential ecotoxicological effects.

*5.3. Field Applications of $Fe^0$-Based Systems*

$Fe^0$-based systems present unprecedented opportunities for wastewater treatment and safe drinking water provision especially in low-income countries, including those in Africa (Table 4) [144,157,191–201]. In fact, the use of $Fe^0$-based systems (e.g., the Bishof Process) for clean water provision has a long history dating back to the 19th century [13,14,202]. The history of $Fe^0$-based drinking water treatment systems is discussed in detail in earlier review papers [17,18,122]. Recently, our research group has proposed the integration of $Fe^0$-based systems in rainwater harvesting systems as a low-cost technology for decentralized drinking water provision, in what is known as the Kilimanjaro Concept [192–194].

**Table 4.** Potential field applications of $Fe^0$-based systems in drinking water and wastewater treatment. The given reference refers to the oldest known application.

| Field of application | Remarks | References |
|---|---|---|
| *A: Safe drinking water provision:* | | |
| 1.Centralized safe drinking water systems | Both filter beds and fluidized beds are used | [168] |
| 2. Decentralized water treatment for small communities | Steel wool is used against radionuclides | [37,154] |
| 3. Household filters against arsenic | Traditional filters are amended with iron nails | [45,46] |
| 4. Household filters against pathogens | Biosand filters are amended with $Fe^0$ materials | [141] |
| 5. Community-scale $Fe^0$-based systems against arsenic | Natural water equilibrates with iron nails and flocs are filtered on gravel | [191] |
| 6. Decentralized rainwater harvesting systems for drinking provision | $Fe^0$ filters are efficient to remediate expected contaminants | [192–194] |
| *B: Wastewater treatment systems:* | | |
| 1. Decentralized domestic wastewater treatment | A $Fe^0$ unit in implemented mostly to remove $PO_4^{3-}$ | [42,195] |
| 2. Wastewater for agriculture and aquaculture | Iron filings are used in filter beds to remove Se | [196,197] |
| 3. Industrial wastewaters/effluents | Sponge iron is used to precipitate Cu and Pb from industrial wastes | [198] |
| 4. *Constructed Wetlands for Wastewater Treatment* | Granular iron is added to reactive materials | [79,199] |
| 5. Urban stormwater treatment | $Fe^0$ is amended to other media to optimize the treatment of runoff water | [200,201] |

$Fe^0$-based systems also have potential applications in domestic and industrial wastewater treatment systems (Table 4). For example, the Harza Process [27,203] has been used to remove Se from agricultural drainage water. Rahman et al. [200] and Fronczyk et al. [201] proposed the amendment of treatment media for runoff infiltration trenches/pits with granular $Fe^0$. In addition, $Fe^0$-based systems have been used to treat both domestic [204] and industrial wastewaters [205,206], including acid mine drainage [24,207]. Available studies suggest that $Fe^0$ can be added as filter media in constructed

wetlands designed to treat urban stormwater and industrial wastewaters [199]. However, a lot remains to be done to further develop and disseminate $Fe^0$-based technologies for wastewater treatment and decentralized safe water provision in developing countries, which such low-cost technologies are most needed. Considering that the bulk of studies are limited to laboratory scale applications, there is need to optimize the $Fe^0$-based systems and evaluate them under field conditions.

## 6. Summary and Conclusions

The corrosion of iron in remediation $Fe^0/H_2O$ systems is an electrochemical process, coupling $Fe^0$ oxidative dissolution to the reduction of water (protons) and to no other available oxidizing agent, including dissolved $O_2$. This is because the universal oxide scale on $Fe^0$ acts as diffusion barrier to dissolved species and a conduction barrier to electrons from the metal body. In other words, water is the sole chemical which can remove electrons from the $Fe^0$ surface. $Fe^0$ oxidation and water reduction must not necessarily occur at the same locality. The spatial separation of oxidative (anodic) and reductive (cathodic) reactions is possible as the metal body allows the free flow of electrons from anodic to cathodic sites. The tendency of $Fe^0$ to give off electrons (Equation (1)) is the same for all $Fe^0$-based materials ($E^0$ = –0.44 V). This makes material selection and characterization critical in designing sustainable $Fe^0/H_2O$ systems.

The need to characterize $Fe^0$ materials in terms of intrinsic reactivity and efficiency is critical for the design and operation $Fe^0/H_2O$ systems, a subject that has been addressed by our research group. Specifically, EDTA and Phen tests were used to characterize the intrinsic reactivity of $Fe^0$ materials. In this regard, the Phen test is considered an affordable and appropriate method that provides a reliable guidance in selecting Fe0 from the large catalog of available $Fe^0$ materials and to control the quality of newly manufactured ones. The efficiency of the $Fe^0/H_2O$ system is characterized using the methylene blue (MB) discoloration method, while other probing agents investigated include orange II, methyl orange and reactive red 120.

The most characteristic issue of remediation $Fe^0$ is the small size (<5 mm) of used materials. Assuming uniform corrosion, the corrosion rates for progressive $Fe^0$ oxidation should be normalized to the individual particles. In other words, expression like mmol year $^{-1}$ should be expressed as mmol year$^{-1}$ particle$^{-1}$ or mmol year$^{-1}$ grain$^{-1}$. The next important issue will be to consider the non-linear kinetics of the corrosion rate such that the service life of a designed system can be deduced knowing the size of used particles and the long-term corrosion rate. Once this is known, considering the expansive nature of iron corrosion would help to design sustainable systems. The choice of the admixing aggregates (e.g., gravel, $MnO_2$, pumice, sand) and the mixing ratios are to be investigated on a case-by-case basis.

A better understanding of the long-term corrosion of relevant $Fe^0$ materials under site-specific conditions is envisioned to ultimately aid in the design of affordable, applicable and efficient remediation $Fe^0/H_2O$ systems. Applications of $Fe^0$-based systems include; (i) a large variety of water treatment systems, (ii) household and small-scale water treatment plants, including rainwater harvesting systems for drinking water supply, (iii) decentralized domestic wastewater treatment, (iv) urban stormwater, agricultural and industrial wastewater treatment, and (v) as filter media in constructed wetlands. Addressing the key knowledge gaps highlighted here, and extending $Fe^0$-based systems to other application domains such as wastewater treatment for agriculture are focal research areas in our group, which brings together collaborators from various countries.

**Author Contributions:** R.H., H.Y., R.T., X.C., M.X., B.K.A., V.C., M.L., N.P.S.-S., A.I.N.-T., N.G.-B., V.R.S.-T., W.G. and C.N. contributed equally to manuscript compilation and revisions. All authors have read and agreed to the published version of the manuscript.

**Funding:** The work is supported by the Ministry of Education of the People's Republic of China through the Program "Research on Mechanism of Groundwater Exploitation and Seawater Intrusion in Coastal Areas" (Project Code 20165037412).

**Acknowledgments:** We acknowledge support by the German Research Foundation and the Open Access Publication Funds of the Göttingen University.

**Conflicts of Interest:** The authors declare no conflict of interest.

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
