# Peer review of "Metallic Iron for Environmental Remediation: Starting an Overdue Progress in Knowledge"

_water, doi:10.3390/w12030641_

Round 1

Reviewer 1 Report

The manuscript summarizes the research on the use of metallic iron for water treatment and remediation. In particular, the authors pinpoint the misconceptions in many previous studies caused by merely focusing on investigation of zero-valence Fe behavior while excluding the role of the oxide coating(scale). The topic is important. The authors emphasize the need for system analysis for Fe material. Because different types of parameters must be considered for different applications with many of these being unclear, the issue seems too complex to resolve in the near future.

The overall structure of this review is good. I would suggest the authors consider the following revision to better support their survey and conclusions.

In the introduction, a table can be added to summarize the type of Fe0 material and its purpose/applications in water treatment. The influencing factors such as morphology, particle size, target contaminants, and field conditions. This will help the readers quickly grasp the current status of the research field reviewed in the manuscript. This can also replace Table 3 in the current manuscript.

In section 4, a table shall be inserted comparing the methods for determining the intrinsic reactivity of Fe0 specimens including EDTA test, Phen test, and tests using MB, orange II, methyl orange and reactive red 120, in terms of the form of Fe analyzed, sensitivity, and limitations.

Author Response

Reviewer 1:

The manuscript summarizes the research on the use of metallic iron for water treatment and remediation. In particular, the authors pinpoint the misconceptions in many previous studies caused by merely focusing on investigation of zero-valence Fe behavior while excluding the role of the oxide coating (scale). The topic is important. The authors emphasize the need for system analysis for Fe material. Because different types of parameters must be considered for different applications with many of these being unclear, the issue seems too complex to resolve in the near future.

Many thanks for this evaluation, and the positive comments!

The overall structure of this review is good. I would suggest the authors consider the following revision to better support their survey and conclusions.

Comment 1:

In the introduction, a table can be added to summarize the type of Fe0 material and its purpose/applications in water treatment. The influencing factors such as morphology, particle size, target contaminants, and field conditions. This will help the readers quickly grasp the current status of the research field reviewed in the manuscript. This can also replace Table 3 in the current manuscript.

Many thanks for this suggestion! We understand the point of Reviewer 1 but is was the aim of the manuscript to present Aqueous Iron Corrosion in its most fundamental aspect before coming back to remediation systems.

We revised by including Table 1 summarizing the history of Fe0 including nature of materials used.

Comment 2:

In section 4, a table shall be inserted comparing the methods for determining the intrinsic reactivity of Fe0 specimens including EDTA test, Phen test, and tests using MB, orange II, methyl orange and reactive red 120, in terms of the form of Fe analyzed, sensitivity, and limitations.

Many thanks for this suggestion! The Phen and EDTA tests, including data obtained are summarized in Section 4 in the manuscript. This is further discussed in Section 5.1 included in the revised manuscript. We are of the view that an additional table with the same information will add limited value to the manuscript.

Reviewer 2 Report

This is an interesting overview about metallic iron application but some aspects should be revised.

The different adsorption isotherms models for iron recovery should be referred.  Also, the differences between each model considered. A more detailed study on the adsorption isotherm can be exposed.

The kinetics also be more explored

Iron provide from waste of other industries should also be considered as a circular economy.

Some references are too old, authors should can found more recent references.

The authors should present some future perspectives of the work.

Author Response

Reviewer 2:

This is an interesting overview about metallic iron application but some aspects should be revised.

Thanks for this evaluation!

Comment 1:

The different adsorption isotherms models for iron recovery should be referred. Also, the differences between each model considered. A more detailed study on the adsorption isotherm can be exposed.

The kinetics also be more explored

We thank the review for the comment, and would like to respond as follows. A detailed discussion of the validity of adsorption isotherms and kinetic models were beyond the scope of the current manuscript. We have considered this and resolved that this is a complex subject that cannot be addressed adequately in the current manuscript. In fact, this is the subject of another manuscript.

Therefore, we revised as follows:

A detailed discussion of the various adsorption isotherm and kinetic models, including assumptions and equations are presented in earlier reviews [109]. A comprehensive review of the limitations associated with the applications of such models to Fe0 systems is the subject of another paper, hence is beyond the scope of the current study. Briefly, the extension of such models to Fe0 materials is problematic for the following reasons: first, unlike adsorbents such as activated carbon whose surface area is readily available for adsorption at the beginning of the experiment (time t0 = 0; Figure 1), Fe0 systems are highly dynamic and equilibrium is rarely achieved in such systems (i.e., rust never rests), while the total reactive sites contributing to contaminant removal is not exactly known (Figure 1).

Comment 2:

Iron provide from waste of other industries should also be considered as a circular economy.

We thank the reviewer for the valid point, and revised the outlook as follows.

‘Besides developing dedicated Fe0 materials for the water treatment industry, scope also exists to use Fe0 material generated as wastes from other industries. Similarly, iron oxide-rich spent sludge from Fe0-based water treatment systems can be used as raw materials in other industries such as the production of pigments (e.g., iron oxide red) (Chen et al., 2015), and even as filter material in the construction industry. This cyclic flow of iron materials between the water treatment industry and other industrial processes could form part of a circular economy.’

We cited the following reference:

Chen, Z., Wang, X., Ge, Q. and Guo, G., 2015. Iron oxide red wastewater treatment and recycling of iron-containing sludge. Journal of Cleaner Production, 87, pp.558-566.

Comment 3:

Some references are too old, authors should can found more recent references.

We have considered the reviewer’s comment. The paper gives a historical review of the Fe0 research field. Thus, the citation of old references, which provide the pioneering work in the field is unavoidable. Moreover, the inclusion of Table 1 on historical applications of Fe0 as recommended by reviewer 1 even requires us to include even older references. However, we have made every effort to include recent references on the subject as evident in the reference list.

Comment 4:

The authors should present some future perspectives of the work.

We agree and would like to respond as follows:

The manuscript has a section on future perspectives (Section 5) which we have further improved by revising as follows:

  • Included section 5.1 highlighting further perspectives
  • Included a brief section on the need to consider Fe0 materials produced as wastes in other industries and then need to develop iron oxide rich sludge into value-added products as part of the circular economy.
  • Revised to indicate that:

‘Addressing the key knowledge gaps highlighted here, and extending Fe0-based systems to other application domains such as wastewater treatment for agriculture are focal research areas in our group, which brings together collaborators from various countries.

Reviewer 3 Report

The review gives an in-depth (critical) literature overview on environmental remediation and water treatment using metallic iron as reactive agent. It provides a thoughtful and detailed review that covers background and both past (somewhat forgotten) and current knowledge on the topic. The paper discusses all the aspects of the use of Fe0/H2O system for water treatment, including the fundamental aspects, chemical aspects, physical aspects, kinetic aspects. It also gives an overview of the common mistakes of past efforts to investigate the use of Fe0/H2O system for water treatment, as well as propose the refinements for future studies on Fe0 remediation.

The paper is well organised, logically sound and the arguments are carefully written to address the topic. The references to previous studies are appropriate and well chosen, with a great number of references by the authors (mainly the author Noubactep C).

My only suggestion to authors would be to re-examine and maybe expand the Keywords list.

Author Response

Reviewer 3:

Comment 1:

The review gives an in-depth (critical) literature overview on environmental remediation and water treatment using metallic iron as reactive agent. It provides a thoughtful and detailed review that covers background and both past (somewhat forgotten) and current knowledge on the topic. The paper discusses all the aspects of the use of Fe0/H2O system for water treatment, including the fundamental aspects, chemical aspects, physical aspects, kinetic aspects. It also gives an overview of the common mistakes of past efforts to investigate the use of Fe0/H2O system for water treatment, as well as propose the refinements for future studies on Fe0 remediation.

Many thanks for this evaluation!

Comment 2:

The paper is well organized, logically sound and the arguments are carefully written to address the topic. The references to previous studies are appropriate and well chosen, with a great number of references by the authors (mainly the author Noubactep C).

Many thanks for these compliments!

Comment 3:

My only suggestion to authors would be to re-examine and maybe expand the Keywords list.

Many thanks for this suggestion, we have added: decentralized water supply, sand filtration, wastewater treatment.

Keywords: Adsorption capacity, decentralized water supply, electrochemical reaction, inconsistent view, sand filtration, wastewater treatment, zero-valent iron.

Round 2

Reviewer 1 Report

The manuscript has been improved with better-organized content, more complete citations and more insightful discussions. The only thing I want to point out is that Figure 1 in the revised version did not show any image as in the initial version.

Author Response

We thank the reviewer for the positive comments, and for drawing our attention to the omission of Figure 1. It appears Figure 1 got corrupted during the revision processes.

We have included Figure 1 in the current revision.

Reviewer 2 Report

The authors answer to all the questions therefore the manuscript should be accepted. 

Author Response

We thank the reviewer for the positive comment.
